# Reducing the Number of Sensors in the Data Glove for Recognition of Static Hand Gestures

Jakub Piskozub * and Pawel Strumillo

Institute of Electronics, Lodz University of Technology, 90-924 Lodz, Poland; pawel.strumillo@p.lodz.pl

* Correspondence: jakub.piskozub@dokt.p.lodz.pl

**Abstract:** Data glove devices, apart from being widely used in industry and entertainment, can also serve as a means for communication with the environment. This is possible thanks to the advancement in electronic technology and machine learning algorithms. In this paper, the results of the study using a designed data glove equipped with 10 piezoelectric sensors are reported, and the designed glove is validated on a recognition task of hand gestures based on 16 static signs of the Polish Sign Language (PSL) alphabet. The main result of the study is that recognition of 16 PSL static gestures is possible with a reduced number of piezoelectric sensors. This result has been achieved by applying the decision tree classifier that can rank the importance of the sensors for the recognition performance. Other machine learning algorithms were also tested, and it was showed that for the Support Vector Machines, *k*-NN and Bagged Trees classifiers, a recognition rate of the signs exceeding 90% can be achieved just for three preselected sensors. Such a result is important for a reduction in design complexity and costs of such a data glove with sustained reliability of the device.

**Keywords:** data glove; gesture recognition; haptic interfaces; piezoelectric sensors; sign language

## 1. Introduction

In the last decade, one can observe a significant development of virtual reality (VR) technology which has resulted in the introduction of systems using this technology to the consumer market. Such systems usually consist of a head-mounted display (HMD), commonly called VR goggles, controllers, and a pair of transceivers. The HMD is a screen, placed at such a distance from the eyes to create the impression of being in the displayed image. This impression is so strong that it has become known as the second reality—a virtual reality. However, a device that creates a near-reality phenomenon should also be expected to interact to further enhance the sense of real-world experience. Currently, the most used human interaction devices are joysticks called wands. They are a kind of combination of the human hand's natural way of interacting with the world around us with existing input devices such as joysticks or computer game pads. For most people, especially youngsters, the control with such a "wand" is widely accepted and in a way natural, because they have been accustomed to using such devices since their childhood [1]. Additionally, the technologies for intuitive interaction with computers have been developed for years, e.g., through computer games. Computer games often require the player to perform several actions at the same time, e.g., moving simultaneously in the game space, controlling the camera, and performing a certain action. Additionally, online games require all these actions to be performed under stress due to action of the competing players and time pressure.

To identify the next stage in the development of human–computer interaction, one can follow the simplified path by which control of modern devices occurs. In order to perform the user's original intention, whether it be a movement in a computer game or the desire to write on the computer keyboard, the user's brain must first translate it into a series of hand movements and then enforce those movements. Thus, any interaction with a computer can be represented as a brain–hand–controller–computer chain [2].

There is already another solution to the problem of controlling a computer, which is the brain computer interface (BCI) [3]. However, it is currently an early stage technology, allowing for the simplest of actions, and is so expensive and non-ergonomic that it will not exist as a commercial solution for several years yet. It can therefore be concluded that the next significant step in the development of control technology will be to eliminate the need to use controller buttons. The greatest benefits of control with a controller that replicates the natural movements of our hand can be achieved when applied to objects with a range of motion similar to that of humans.

In the era of growing importance of telecontrol and breakthroughs in robotics, it also becomes necessary to develop a device that would allow an intuitive control of robotic manipulators while maintaining the required precision of their operation. However, for this to be possible, it is necessary to recreate the motor-nervous system of the hand so that the operator can freely perform operations and experience their consequences. This is possible with the use of devices that can track the user's hand movements. A device that allows direct measurement of finger movements was developed and described by Sandin et al. [4] and was named data glove. The form of the glove makes the use of this device ergonomic and does not interfere with the tested movements. This allows for natural movement, which is very important when testing all kinds of gestures and sign language.

This paper describes a study using a designed data glove. The research focuses on two main objectives:

1. To test the effectiveness of the constructed glove in recognizing static hand gestures;
2. To investigate the possibility of reducing the number of sensors used in the recognition static hand gestures.

To the best of our knowledge, no research has yet been conducted on the feasibility of reducing the number of detectors in structures of similar purpose, and no research with publicly available results has been conducted on the Polish Sign Language alphabet using 10 flex sensors.

Reducing the number of sensors, and thus the size of the feature vector, will simplify not only the construction of the glove itself but also the classification process. A simpler design and a reduction in the necessary computational power could improve the usability of such a data glove. Furthermore, less complex data gloves that can be used as sign-language-to-speech translators may be less expensive and may be more widely accepted.

*Sign Language Alphabet and Stance of Sign Language Speakers*

It is important to draw a distinction between sign language as such and fingerspelling. Every sign language has its own grammar and vocabulary. However, the syntax of many of them is based on signs that are equivalents of whole words or phrases in a spoken language. The fingerspelling alphabet, on which the Polish version of the gestures presented in this article are based, is a supplement to the sign language. Fingerspelling is also used to make signs corresponding to counters, numbers, or punctuation. These sign language signs are made solely with the hands, two for British Sign Language and one for Polish Sign Language, among others, and do not require facial expressions or poses. Spelling in sign languages is most used when pronouncing proper names such as first names or street names or when learning other sign language signs [5].

The stance of the sign language community towards sign-language-to-speech glove-translators should also be made clear here. Many of them have a negative attitude towards this type of device. This is mainly due to bulkiness and low efficiency in sign language recognition [6]. While the first of these obstacles can be overcome by striving for miniaturization of such devices, for example, by reducing the number of sensors used (as described in this paper), the low effectiveness is due to the inability of the gloves to capture other important factors of sign language such as facial expression or pose [7]. However, although the data glove may not be the ultimate solution to the problem of translating sign language into speech, mainly due to the reasons stated above, it seems it is currently an excellent device for recognizing hand movements. The readings obtained with them and the databases

created from them will allow for the development of more complete and accurate training sets for sign and sign word recognition algorithms, thus increasing their effectiveness in future implementations.

It should also be noted that the gestures referred in this work are not letters of the Polish Sign Language manual alphabet. It was assumed that the main difference between letters is their main, static form. However, such an assumption makes it necessary to classify the handshapes as gestures rather than letters of PSL due to lack of dynamic stages of presentation of individual characters.

The next section describes articles on applications of data gloves in gesture recognition and sign languages. Section 3 describes the design of the glove and the research methodology, and the results are presented in Section 4. The article concludes with a summary of the results in Section 5 and a brief discussion undertaken in Section 6.

## 2. Related Work

Analyzing the human hand as a mechanical system, there are a total of 27 degrees of freedom [8]. Considering individual human motoric coupling, the human hand is a very complex mechanical manipulator. In addition to the complex muscular and skeletal system, the hand (apart from the central nervous system), is the most innervated part of the human body with a significant cortical representation. The human brain devotes considerable resources to processing the signals of the sense of touch. This is a crucial and complex task, to the extent that it requires the synchronous use of both hemispheres of the brain. This is due to the need to provide feedback for all hand actions [9].

In [10], the authors reviewed numerous articles published in the past year about the data glove, focusing on their capabilities in recognizing sign language (SL) signs or gestures. It is important to point out the differences between SL signs, which can be divided in terms of the dynamics of a given gesture or sign. Static signs are presented in a sequence, and there is a natural suspension point in each sign. Dynamic gestures, on the other hand, are challenging due to the lack of such a point. In addition, with SL, quite like speech, the spacing between each sign is important so that it can be understood. Differences also occur at the level of semantics. In the case of SL, all its signs have a separate and independent meaning. Gestures can also be an independent form of communication, but there is a specific group of them whose role is only to reinforce another form of communication. Additionally, there is a group of gestures that have a purely mechanical function.

The documented designs of the data gloves can be divided into two main groups according to the principle of measurement (see Figure 1). The devices with indirect measurement method use a medium to read the position of the hand, and devices using the direct method are attached to it and thus track its movements.

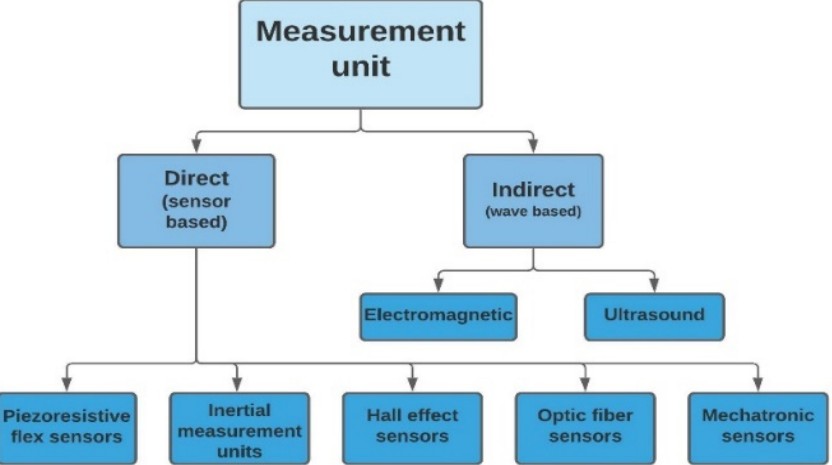

**Figure 1.** Classification of hand movement measuring devices according to their principle of operation.

Indirect measurement methods include electromagnetic waves and sound waves [8,9,11]. The most popular indirect method is the one that uses cameras to track a hand using machine vision techniques. A popular device of this type is Ultraleap's Leap Motion, recently used to recognize 100 Arabic Sign Language gestures [11]. However, image-based hand tracking is subject to limitations due to the measurement method itself. The camera cannot capture movements that take place outside its field of view and those that are obscured by another object. Many everyday gestures or hand movements such as shaking hands may cause the skeleton tracking algorithm to be unable to map the position of the hand. This problem does not occur with hand tracking using the data glove [12,13].

The second group of devices are those that use sensors placed directly on the hand. These devices are called data gloves because of their glove-like shape. They can be classified according to the type of sensors used. One of the most popular types of sensors, probably due to its relatively low price and ease of application, are piezoresistive sensors [10].

Maitre et al. [14], using a self-built glove, investigated the behavior of the hand while holding seven everyday objects in the palm of the hand and made a successful attempt to classify these grasps based on the collected data.

Pezzouli et al. [15] conducted an experiment using a similar glove and tested the effectiveness of the device in recognizing 27 gestures and compared several methods for their segmentation and classification. Each glove has 10 deflection sensors and one Inertial Measurement Unit (IMU) for tracking hand orientation. The authors also conducted an experiment using a publicly available database containing readings obtained from a similar glove while performing sign language gestures. In both experiments, the team obtained high classification accuracy of 90% for most of the tested methods.

Mummadi et al. [16], on the other hand, presented a comparison of several methods for recognizing 22 static French sign language gestures. Data obtained from a built glove, employing five IMUs, were classified with the use of algorithms based on Random Forests, Naive Bayes Classifier and Multi-Layer Perceptron.

Zhang et al. [17] proposed a glove that is also based on piezoresistive sensors but in combination with an electromyograph and inertial sensors placed on the glove. The effectiveness of the solution was tested on 10 gestures, of which 5 were dynamic gestures, and achieved a recognition rate of 89.3% for static gestures and 76.6% for dynamic gestures.

Significant advances have also been made in the field of sensors themselves. Xie et al. [18] have developed a method for fabricating triboelectric nanogenerators for textronic applications. The nanogenerators are highly flexible and can be freely trimmed. Their electrical properties allow them to be used as active deflection sensors. Such a potential application was tested by measuring the voltage signals generated by the sensors when individual fingers were bent. A preliminary analysis of the obtained results shows the potential validity of this solution in data glove-type devices. A similar method and results were also reported in [19].

Huang et al. [20] presented sensors using graphene technology. A layer of reduced graphene oxide (RGO) was sputtered onto a piece of double-covered yarn (DCY) giving it piezoresistive properties. The performance of the sensors was tested on a set of 10 gestures representing digits from 0 to 9 and 9 dynamic Chinese Sign Language characters using artificial neural networks (ANN) [21] and dynamic time warping (DTW) method. The achieved recognition performance was 98.3%.

A different method of tracking finger movement was presented by Rinalduzzi et al. [22]. Coils were mounted on the fingertips of the glove, emitting an electromagnetic field at a different frequency for each of the seven transmitters. To function, such a solution needs receivers placed in three planes. The developers presented a wall made for this purpose, in whose space a gloved hand moves. This approach precludes the use of the device in open spaces which is one of the main drawbacks of using vision-based methods for gesture recognition.

In [23], a method of applying the sensors directly to the surface of the material is presented. The resistive fingertip deflection sensors on which the glove performance is based were fabricated using the direct indium–gallium eutectic writing method.

The software part of gesture recognition problem can be divided into two parts: detection and classification. While neural networks and machine learning algorithms can often achieve high classification performance, the need to capture the beginning and end moments of a given gesture poses a significant problem for devices serving as SL translators. Many systems developed to date have not been able to recognize dynamic SL characters, and none have operated in real time outside of laboratory environment. This is due to the difficulty of segmenting readings from such a glove. Separating sign language characters from each other is still a difficult challenge for the currently developed algorithms [10].

An original segmentation approach that is based on a scalar called Gesture Progression Scalar (GPS) was proposed by Lee and Bae [23]. The GPS is determined by the dynamics of resistance changes in 10 sensors. Based on its values, it is possible to determine the moment at which a given gesture starts and ends. It is also worth mentioning that both presented algorithms were tested on a dedicated glove without any inertial sensor, which is a typical approach for dynamic gesture recognition. The ANN-based recognition algorithm was tested on 10 American SL characters.

The final step in the gesture recognition pipeline is gesture classification. Machine learning algorithms are often used for gesture classification. Their use allows researchers to obtain fast classification at the cost of longer learning times [10]. ANN was used by Lee and Bae [23] and Huang et al. [20]; machine learning algorithms such as Support Vector Machine [10,18–24] or $k$-Nearest Neighbors [21–26] and Hidden Markov Models [27–30] are equally often used and juxtaposed.

## 3. Materials and Methods

The glove was made of materials and components generally available on the consumer market. The main objectives of the design were to allow free movement of the sensors while maintaining their position in relation to the finger and the possibility of adjusting the glove to the hand of the subjects. Due to the phenomenon of cracking of the connectors of the sensors used, they were replaced with crimped Amphenol FCI connectors, which additionally enabled quick replacement of most sensors in case of damage or wear.

### 3.1. Hardware

The glove used in the experiment was made of 10 piezoresistive flex sensors: 4 with a length of 112 mm and 6 shorter ones with a length of 73 mm. They were placed on the glove in two rows on each of the five fingers: the first at the level of metacarpophalangeal joints and the second covering interphalangeal joints. The choice of sensor placement assumed that independent flexion of the distal interphalangeal (DIP) joints does not occur in sign languages and that it does not belong to any group of gestures used in everyday life. Additionally, only a small number of people can perform such a flexion naturally.

The sensors were attached to the textile surface of the glove using Velcro. The motion path of the sensors was stabilized by 3D-printed guides which are also attached to the glove using Velcro. Movable sensor mounts allowed us to adjust their placement and motion path to the user's finger spacing. In this way, the axis of the sensor always coincided with the finger axis during measurements, making them as accurate as possible. A photograph of the glove is shown in Figure 2. The sensors were connected to the measurement unit using conductive threads embedded in the glove. The ends of the thread were attached to the sensors with detachable connectors so that the sensors could be easily replaced. The other end of the thread was soldered to the PCB of the measurement system. On the PCB, there are also 15 kΩ resistors to which piezoresistive sensors are soldered so creating a voltage divider.

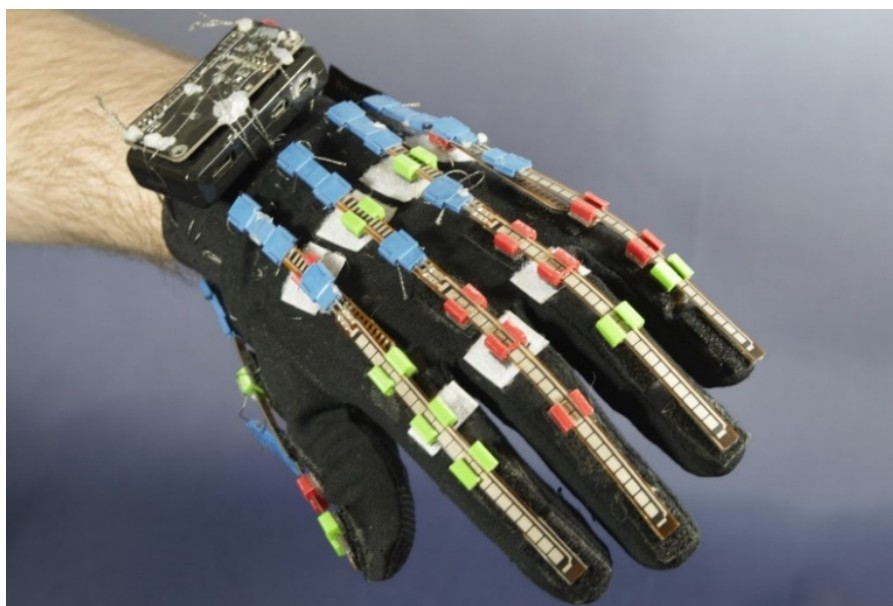

**Figure 2.** The glove used in the study.

Linearity of the resistance characteristics of the sensor with respect to the bending angle was confirmed with a simple experiment measuring the resistance of the sensor in the bending range from 0° to 90° in 5° steps; it was carried out using Axiomet AX-8455 multimeter manufactured by Shanghai Yi Hua V&A Instrument Co. in Shanghai, China.

The measurements of glove sensors are controlled by a microcomputer Raspberry Pi Zero W, which is a single-board computer running on a Broadcom BCM2835 1 GHz processor with 512 MB of built-in RAM. It consists of 4 ADS1115 16-bit analog-to-digital converters working at 860 Hz sampling rate (see Figure 3).

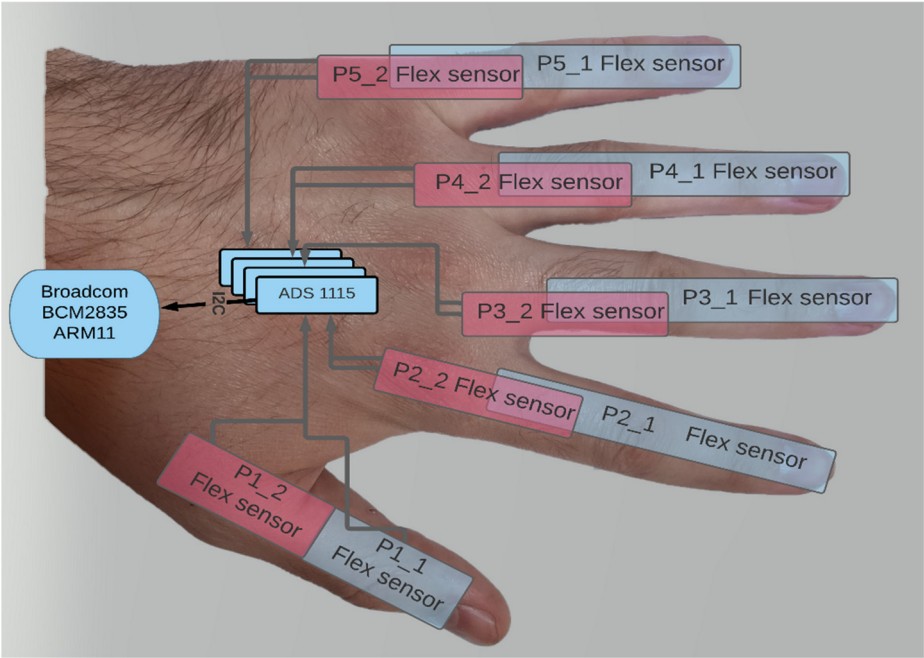

**Figure 3.** Block diagram of the glove.

The operation of the glove was controlled by the Linux operating system in a distribution designed for embedded systems with the program controlling the operation of the acquisition system written in Python.

### 3.2. Database Generation

A special measurement procedure was applied to collect the measurement data from the data glove. The script implementing this procedure controls the readings of 10 sensors in a continuous mode, enabling quick diagnostics of the sensors' operation. After a subject assumed the final, static position of a gesture shown in Figure 4, the script saves 10 consecutive readings of sensor values to a csv file that was automatically created at the start of the measurement.

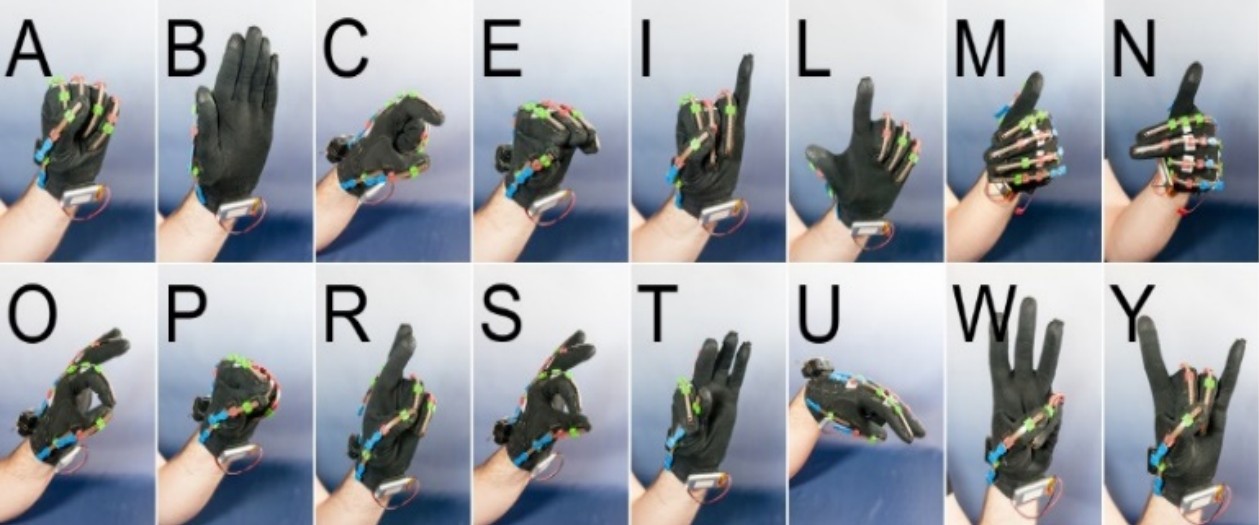

**Figure 4.** Gestures based on static letters of Polish Sign Language alphabet.

There were 15 subjects aged from 23 to 68 who volunteered in the experiments. Five women and ten men took part in the trials. Three of the subjects were left-handed. All subjects had an unrestricted range of hand movement necessary for the study. They had similar hand size ranging from 18 to 21 cm measured from the wrist to the tip of the middle finger, with no known musculoskeletal disorders. None of the subjects had also been exposed to sign language prior to the study.

The tests of the data glove were approved by the Institutional Ethics Committee of Technical University of Lodz (No. 1/2021, date of approval: 28 October 2021). Before starting the measurement, subjects were instructed about the course of the measurement (shown in Figure 5) and were familiarized with 16 gestures based on letters of the Polish Sign Language fingerspelling alphabet, which they were about to present. Each participant performed each of 16 static gestures 10 times. After each repetition, the hand was relaxed for a few seconds, which resulted in slight differences between repetitions.

The measurements of each subject were taken in a natural sitting position, with the elbow resting on a surface and the forearm gently deviating from the vertical. Finger placement in the natural relaxed position was also measured in this position. In total, the database consisted of 25,550 readings of the sensors.

The nominal resistances of the sensors differed even relative to sensors from the same production batch. The reading distribution chart shown in Figure 6 presents the median readings and outliers. The formation of outliers was caused by an imperfection in the way the sensors were connected to the conductive threads unnoticeable at the time of the study. This minor flaw was corrected, while the outliers remained in the dataset on purpose, reflecting an imperfection in the actual functioning of the device. In all the presented results, we used the data with outliers for training the classifiers.

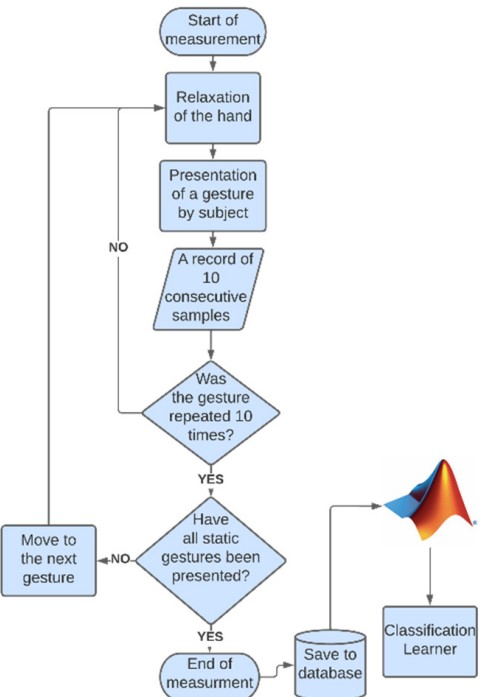

**Figure 5.** The methodology flowchart.

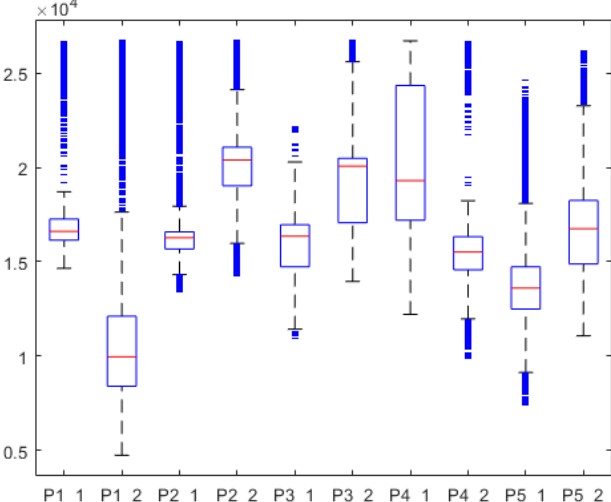

**Figure 6.** Distribution of measurements data.

## 4. Results

Of the many existing sign languages, it was decided to conduct the research based on the Polish Sign Language. In the case of static characters, it is not important which hand they are made with. Therefore, for practical reasons, the left hand was chosen.

All classification processes were performed using a five-fold cross-validation technique.

For the standard *k*-NN algorithm methods, the k parameter was set to 1. For the *k*-NN in Random Spaces (*k*-NNiRS) method, the number of subspaces was chosen automatically by the software and was equal to about half of the total number of data dimensions.

The goal of the experiment was to investigate the possibility of reducing the number of sensors while losing as little gesture recognition performance as possible.

For this purpose, the Maximum Relevance-Minimum Redundancy (MRMR) method was used. It is an iterative method of determining the most relevant features of the dataset under study. The score for each variable is determined as the quotient of significance and redundancy against previously tested variables. Relevance is calculated using

F-statistic between the feature and the expected class, while redundancy is calculated as the average Pearson correlation between the variable under study and the variables checked in previous iterations [31].

Once the sensor hierarchy presented in Figure 7 was determined, data classification was performed using six classifiers. Almost all relevance metrics identified sensors P3_2, P1_2 and P2_2 as the three most relevant. The least significant sensor according to all selection methods was P4_2.

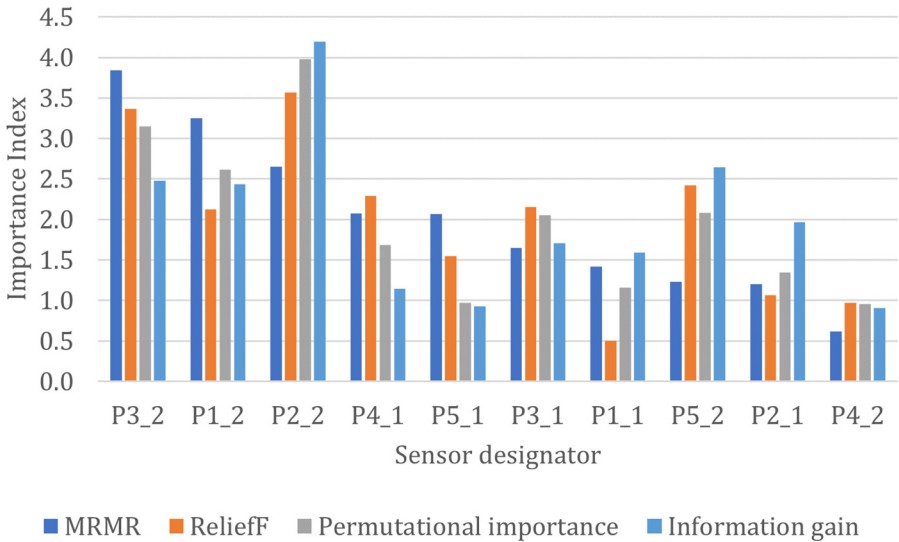

**Figure 7.** Importance of specific sensors.

The effectiveness of each algorithm was calculated using Equation (1)

$$\text{Performance} = \frac{\text{TPR}}{\text{TPR} + \text{FPR}} \times 100\% \tag{1}$$

where TPR means True Positive Rate and FPR stands for False Positive Rate.

The effectiveness of each algorithm for a limited number of sensors is shown in Figure 8.

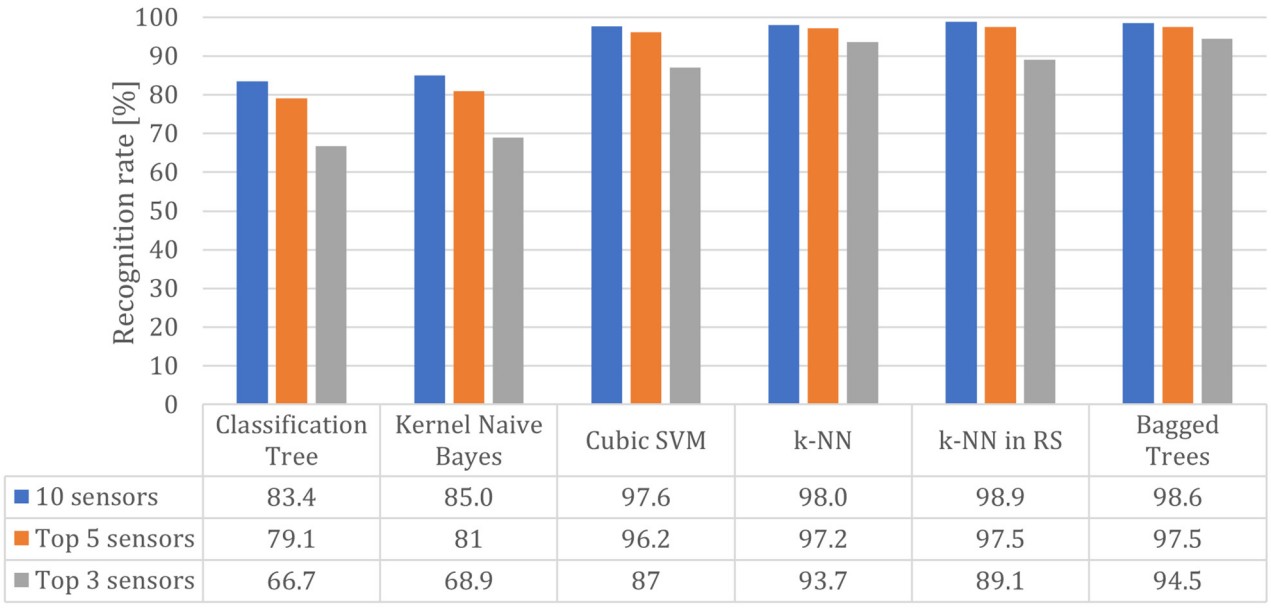

| | Classification Tree | Kernel Naive Bayes | Cubic SVM | k-NN | k-NN in RS | Bagged Trees |
|---|---|---|---|---|---|---|
| 10 sensors | 83.4 | 85.0 | 97.6 | 98.0 | 98.9 | 98.6 |
| Top 5 sensors | 79.1 | 81 | 96.2 | 97.2 | 97.5 | 97.5 |
| Top 3 sensors | 66.7 | 68.9 | 87 | 93.7 | 89.1 | 94.5 |

**Figure 8.** Classification efficiency as a function of the number of sensors used.

The average gesture recognition performance of each tested algorithm from all conducted experiments is shown in Table 1.

**Table 1.** The average effectiveness of each of the 6 classifiers tested.

| Algorithm | Average Recognition Rate [%] |
|---|---|
| Classification Tree | 76.7 |
| Kernel Naive Bayes Classifier | 74.6 |
| Cubic SVM | 91.5 |
| *k*-NN | 93.7 |
| *k*-NN in RS | 91.5 |
| Bagged Trees | 93.4 |

The highest level of correct classifications was achieved by the *k*-NNiRS classifier when working on all 10 sensors. A slightly lower recognition rate was obtained for the Bagged Trees method. The BT method, in contrast to *k*-NNiRS, maintains a performance above 94% regardless of the number of sensors and their placement (Figures 8 and 9). To verify how reducing the number of dimensions in the feature vector affects the efficiency of distinguishing individual characters, the confusion matrix was built for the algorithm with the highest mean efficiency.

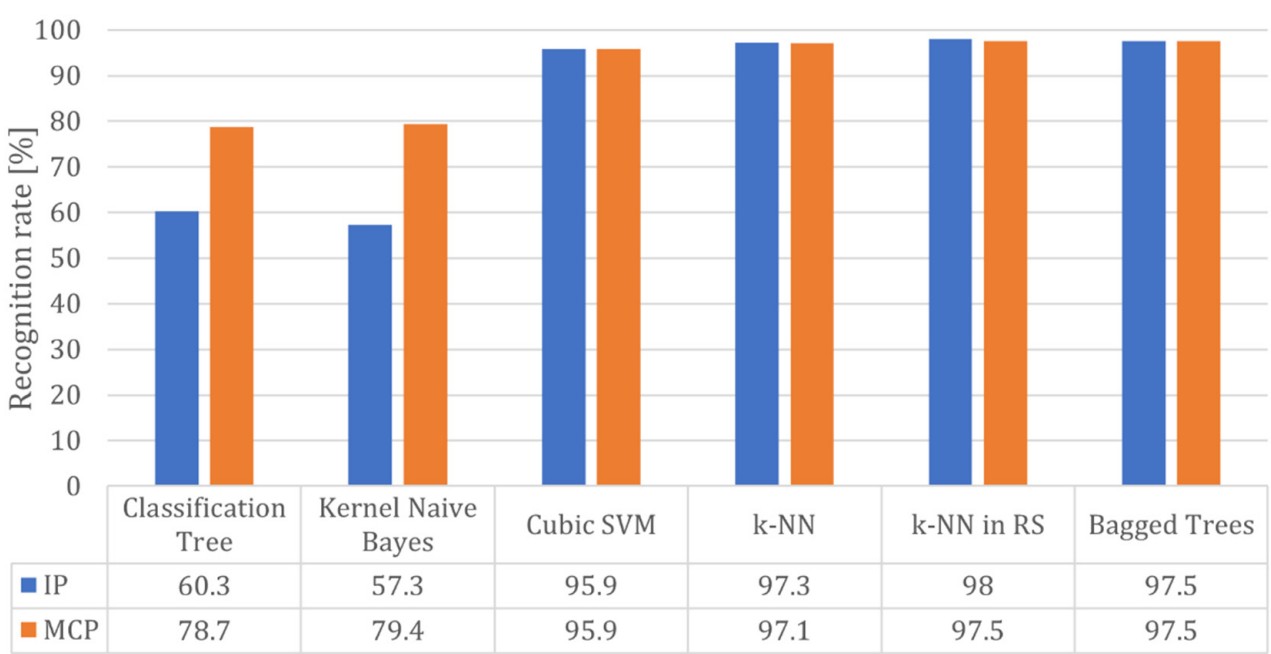

**Figure 9.** Classification performance as a function of the row of sensors used.

Additionally, we re-run the classification process ignoring data from the first interphalangeal (IP) and then the second metacarpophalangeal (MCP) row of sensors to check how placement over specific joints impacts the recognition rates.

The obtained results allow us to conclude that sensor placement over specific joints does not affect the performance of most classifiers. For the Classification Tree and Naive Bayes Classifier, the placement on the metacarpophalangeal joints achieving a 20% improvement in recognition rate.

For the three data classifiers with the highest average performance, the dependence of classification performance on the dimensionality of the feature vector was also tested. In this case, components of the feature vector were sequentially included into the classification process starting with those with the highest importance index as determined by the MRMR method.

In terms of the *k*-NNiRS method, for the full ten-component feature vector, a gesture recognition rate of 98.9% was achieved, while for the three-component feature vector, the recognition rate dropped to −89.1%. As the dimensionality of the dataset decreased, there was a decrease in recognition performance for all characters. However, a several-fold increase in False Negative Rates for a few characters, i.e., "W", "O", "U", "R" and the resting state was observed. The largest drop in recognition performance for the W", "O" and "U" gestures is most likely due to the exclusion of the sensor at the thumb's interphalangeal joint from the classification process. Without this information, the algorithm failed to distinguish whether the thumb was fully flexed or whether the only flexion occurred at the interphalangeal joint and the thumb remains straight. The decrease in performance in recognizing a gesture based on the letter "R" is due to the disabling of the P2_2 sensor at the interphalangeal joint of the index finger. Even though the range of motion necessary to record movement of the fingers at the metacarpophalangeal joint bringing the fingers closer and further apart is theoretically unattainable for the sensors used placed in this plane, the P2_2 sensor nevertheless recorded the overlap of the index and middle fingers that make up the "R" sign. The most poorly recognized sign using a three-component feature vector is the sign based on the letter "W." The cause of this is the similarity of this gesture to many others in the form of straightened fingers 2–4 and a bent thumb and little finger. In addition, for many people, this positioning of the fingers is extremely unnatural, which not always resulted in the correct performance of this gesture. Table 2 shows the confusion matrix obtained for character classification using the most effective *k*-NNiRS algorithm, with a reduced number of sensors used.

**Table 2.** Confusion matrix for *k*-NNiRS method, while *k* = 1, obtained with feature vector limited to three sensors (expressed as a percentage).

| | a | b | c | e | i | l | m | n | o | p | r | s | t | u | w | y | rest | TPR | FPR |
|---|---|---|---|---|---|---|---|---|---|---|---|---|---|---|---|---|---|---|---|
| a | 95.2 | 0.6 | 1.0 | 0.0 | 1.0 | 0.1 | 0.0 | 0.1 | 0.1 | 0.3 | 0.1 | 0.1 | 0.1 | 0.3 | 0.5 | 0.5 | 0.1 | 95.2 | 4.8 |
| b | 0.4 | 90.8 | 0.9 | 0.6 | 0.1 | 0.5 | 0.3 | 0.1 | 0.6 | 0.5 | 1.1 | 0.6 | 0.8 | 1.4 | 0.9 | 0.0 | 0.4 | 90.8 | 9.2 |
| c | 0.4 | 0.6 | 88.1 | 1.0 | 0.5 | 1.2 | 0.4 | 0.4 | 0.8 | 1.8 | 0.8 | 1.0 | 0.6 | 0.6 | 0.8 | 0.5 | 0.4 | 88.1 | 11.9 |
| e | 0.1 | 0.8 | 0.5 | 92.5 | 0.1 | 0.6 | 0.6 | 1.2 | 0.2 | 0.4 | 0.2 | 1.4 | 0.6 | 0.5 | 0.2 | 0.2 | 0.1 | 92.5 | 7.5 |
| i | 0.5 | 0.2 | 0.5 | 0.0 | 95.9 | 0.5 | 0.0 | 0.1 | 0.5 | 0.3 | 0.0 | 0.0 | 0.3 | 0.3 | 0.3 | 0.3 | 0.4 | 95.9 | 4.1 |
| l | 0.4 | 0.4 | 1.1 | 1.0 | 0.6 | 92.2 | 0.1 | 0.6 | 0.1 | 1.3 | 0.1 | 0.1 | 0.2 | 0.3 | 0.1 | 1.0 | 0.5 | 92.2 | 7.8 |
| m | 0.0 | 0.1 | 0.1 | 1.9 | 0.0 | 0.3 | 92.9 | 1.4 | 0.3 | 0.1 | 0.1 | 0.6 | 0.8 | 0.3 | 0.4 | 0.1 | 0.6 | 92.9 | 7.1 |
| n | 0.0 | 0.4 | 0.4 | 1.6 | 0.0 | 0.2 | 1.9 | 87.8 | 1.7 | 0.6 | 0.5 | 2.2 | 0.9 | 0.8 | 0.5 | 0.1 | 0.5 | 87.8 | 12.2 |
| o | 0.4 | 0.4 | 1.6 | 0.5 | 1.0 | 0.1 | 0.6 | 1.4 | 84.3 | 1.1 | 1.4 | 1.5 | 0.6 | 1.8 | 1.6 | 1.0 | 0.6 | 84.3 | 15.7 |
| p | 0.5 | 0.4 | 1.4 | 0.8 | 0.1 | 1.8 | 0.3 | 0.4 | 0.7 | 89.6 | 0.8 | 0.7 | 0.6 | 0.4 | 0.7 | 0.3 | 0.5 | 89.6 | 10.4 |
| r | 0.1 | 1.1 | 0.3 | 0.3 | 0.1 | 0.5 | 0.3 | 0.6 | 1.6 | 0.3 | 86.2 | 0.5 | 0.6 | 3.3 | 2.8 | 1.0 | 0.5 | 86.2 | 13.8 |
| s | 0.4 | 0.6 | 0.5 | 1.4 | 0.1 | 0.3 | 0.5 | 1.4 | 2.1 | 0.6 | 0.2 | 89.6 | 1.4 | 0.5 | 0.3 | 0.1 | 0.1 | 89.6 | 10.4 |
| t | 0.0 | 1.0 | 0.5 | 0.4 | 0.1 | 0.3 | 1.0 | 0.6 | 0.7 | 0.9 | 1.5 | 1.0 | 88.2 | 1.9 | 0.8 | 0.3 | 0.8 | 88.2 | 11.8 |
| u | 0.1 | 1.4 | 0.8 | 0.3 | 0.5 | 0.1 | 0.5 | 1.0 | 1.4 | 0.1 | 2.8 | 0.7 | 1.4 | 83.8 | 2.4 | 2.1 | 0.6 | 83.8 | 16.2 |
| w | 0.3 | 1.3 | 0.4 | 0.3 | 0.6 | 0.1 | 0.4 | 0.8 | 2.3 | 0.8 | 3.2 | 0.4 | 0.7 | 2.7 | 82.2 | 2.6 | 0.9 | 82.2 | 17.8 |
| y | 0.9 | 0.1 | 0.6 | 0.0 | 1.1 | 1.2 | 0.1 | 0.3 | 0.7 | 0.8 | 0.7 | 0.1 | 0.3 | 1.7 | 1.6 | 89.3 | 0.5 | 89.3 | 10.7 |
| rest | 0.3 | 1.0 | 0.8 | 0.2 | 0.2 | 0.3 | 0.7 | 0.9 | 1.0 | 1.2 | 1.0 | 0.3 | 1.6 | 0.9 | 2.0 | 1.7 | 85.6 | 85.6 | 14.4 |
| | a | b | c | e | i | l | m | n | o | p | r | s | t | u | w | y | rest | TPR | FPR |

TRUE CLASS (vertical axis label) — PREDICTED CLASS (horizontal axis label)

To test the recognition performance of characters not relying on the P3_2, P1_2 and P2_2 sensors, the average readings for two other very similar characters were examined. The characters "O" and "S" differ in the flexion of only one joint. As can be seen in Figure 10, the average readings for the two characters clearly point to the P1_1 sensor taking different readings.

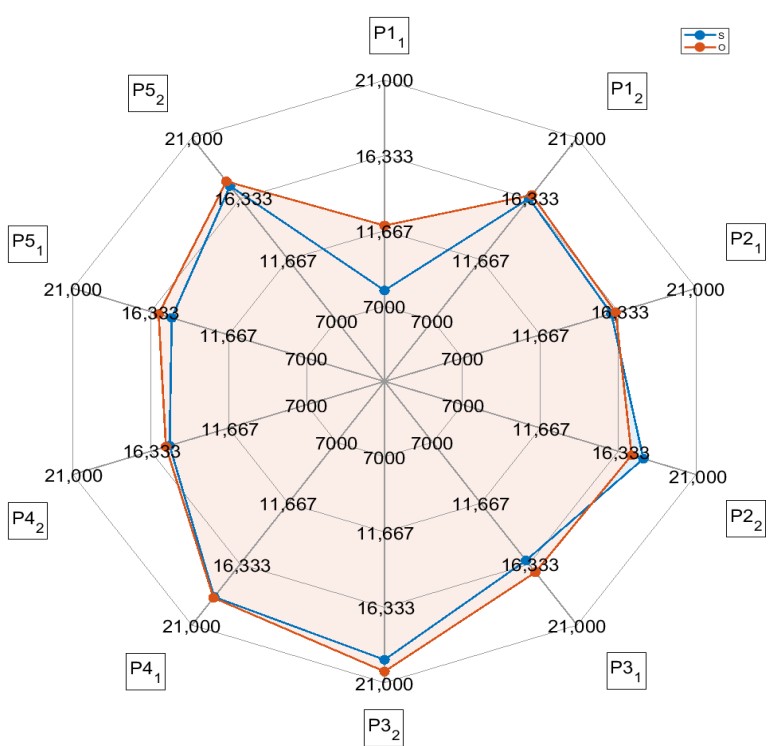

**Figure 10.** The average signal values of each sensor for the gestures corresponding to letters "O" and "S".

## 5. Discussion

An interesting feature of the PSL is that it has diacritics that are dynamic versions of corresponding letters. This feature is important for further research involving recognition of dynamic signs.

The initial phase of the study focused on testing the overall performance of the data glove. A significant challenge was the recognition of gestures with similar finger arrangements. As shown in Figure 10, the differences in two similar gestures captured by the presented glove are visible to the naked eye. The finger placements for both gestures are very similar to each other and differ only in the flexion of one joint.

Importantly, reducing the dimensionality of the feature vector according to the designated sensor importance did not result in a significant decrease in the classification performance of the recognized characters (Figure 11).

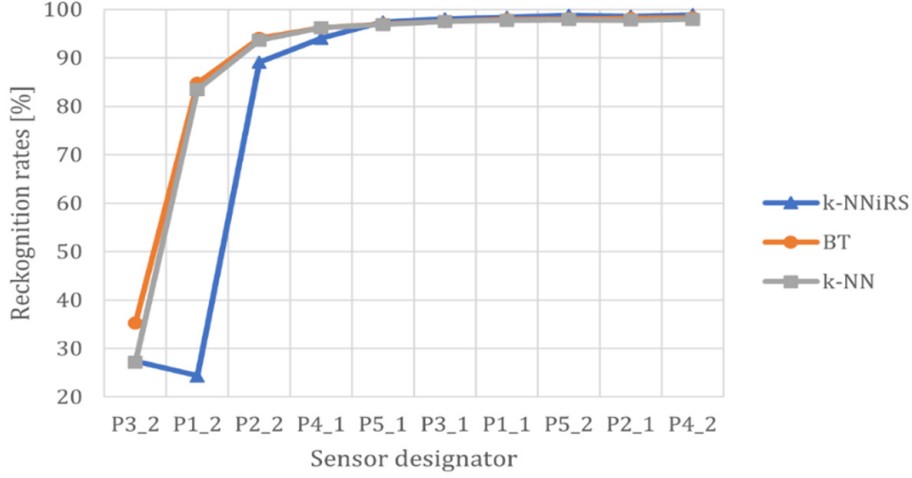

**Figure 11.** Graphs of gesture classification performance by algorithms with the highest average performance against the number of sensors.

Reducing the dimensionality to five features resulted in a 97.5% effectiveness for *k*-NNiRS and BT algorithms.

Although the efficiency of the *k*-NNiRS algorithm turned out to be the highest of all classifier for 10 sensors, it decreased most rapidly as the dimensionality of the feature vector was reduced. This is due to the main assumption of the algorithm, which includes self-reduction method of the feature vector to a randomly selected few of its components. This explains the apparent decrease in classification efficiency of the *k*-NNiRS algorithm, as seen in Figure 11, while the BT and *k*-NN algorithms achieve efficiency of 85%. The *k*-NNiRS algorithm in this situation uses only a one-dimensional feature vector.

Table 3 compares the work presented in Related Work with the results presented in this paper. It should be noted that the solutions for recognizing dynamic gestures and static gestures cannot be directly compared due to the different methods used. The main contribution of study is the method of reducing the number of sensors needed to recognize static gestures based on Polish Sign Language to 3 out of 10 sensors. As can be seen in Table 3, no device presented in the previous works used a device with such a small number of sensors.

**Table 3.** Comparison of the results presented in this paper to articles in Related Work.

| Author | Ref. Num. | Kind of Gestures | Highest Performance | Number of Sensors | Method | # Gestures | # Users |
|--------|-----------|------------------|---------------------|-------------------|--------|------------|---------|
| Maitre et al. | [14] | static, based on daily activities | 99 | 15 piezoresistive | *k*-NN/RF | 8 | 9 |
| Pezzouli et al. | [15] | dynamic, based on Australian Sign Language | 99.7 | 10 piezoresistive + 1 IMU | RF | 27 | 5 |
| Mummadi et al. | [16] | static, based on French Sign Language | 91 | 5 IMUs | RF/ANN | 22 | 17 |
| Zhang et al. | [17] | dynamic, based on daily activities | 89.34 | 5 piezoresistive + Myo armband | LSTM | 10 | 10 |
| Xie et al. | [18] | static, based on daily activities | n/a | 5 custom made spiral steel wire based | n/a | 8 | 1 |
| Huang et al. | [20] | dynamic, based on Chinese Sign Language | 98.3 | 10 RGO piezoresistive | ANN | 9 | 2 |
| Lee and Bae | [23] | dynamic, based on American Sign Language | 100 | 10 GaIn piezoresistive | LSTM | 11 | 1 |
| This work | n/a | static, based on Polish Sign Language | 98.9/**94.5** | 10/**3** piezoresistive | *k*-NNiRS/ *k*-NN | 17 | 15 |

An efficiency of 95.1% was achieved by BT and *k*-NN classifiers for the three-component feature vector (Figure 8). A more than ten-fold increase in FPR was observed for several hand positions including those corresponding to letters "W", "O", "U" and "R". The gestures corresponding to letters "W" and "U" were most often confused with the gesture corresponding to letter "R". This is most likely because all these gestures require the index and middle finger to be fully straightened, while overlapping these fingers while performing the gesture corresponding to letter "R" requires detection of a movement that is not performed in the plane of movement of the sensor. The decrease in TPR for the letter "O" can be explained in a similar way. It relies mainly on the index finger and thumb, with middle finger straight.

The exclusion from the classification process of the sensor located above the interphalangeal joints of the index finger, which flexes the most during this gesture, may result in a false prediction of the gesture. Nevertheless, the performance obtained can be considered satisfactory and promising, as it remains above or close to 90% for most gestures. This is a result worth noting because *k*-NN and BT are classifiers with short training time and low computational complexity.

The proposed reduction in the number of sensors results in a decrease in the recognition efficiency of the tested gestures to 93–95%. Table 2 shows that the largest contribution to these errors comes from misrecognized gestures corresponding to letters "W" and "U".

In its current form, the system can be used for simple fingerspelling interactions. It would only allow the user to provide simple information or to translate proper names to people without knowledge of sign language or write on a computer using the manual alphabet. A version of the system having a set of 10 flex sensors and inertial sensors will also be an excellent tool for creating freely available databases. The aim of a future study is to develop a system consisting of two hand overlays based on as few sensors as possible. A solution could be a system that combines hand motion recognition using a glove and a smartphone camera to track mouth movements, pose, and facial expressions of the speaker.

## 6. Conclusions

For the recognition of hand gestures drawn from PSL static letters, it is not necessary to use 10 sensors monitoring the flexion of all members of each of the 5 fingers to achieve classification performance exceeding 97%.

For the problem of recognizing the tested static hand gestures, three sensors at the level of the metacarpal joints are most relevant: on the middle finger, on the thumb, and on the index finger.

The classification process based only on these three values achieves a performance of 95% for the k-NN method for $k = 1$. It is also the most resilient to reductions in the dimensionality of the input data and to interference from hardware defects.

Despite the exclusion of key features for some characters, the recognition rate for most characters is sustained at a level close to 90%.

The above conclusions apply only to the studied case of the static gestures. The next step in the research will be to test the validity of reducing the number of sensors in the dynamic hand gestures of an actual PSL manual alphabet recognition task, also performed in sequence. Further studies will solicit the participation of PSL users when the feasibility of using fewer sensors in recognizing actual PSL letters is confirmed by preliminary results. With their participation, the plan is to continue building a database of glove signals when using PSL. If the above proposal is positively received by the sign language community, the possibility of reducing the number of sensors in data glove-type devices used as sign language translators can bring significant benefits to both designers and potential users. We believe that using fewer sensors would not only simplify the design and thus reduce the number of unreliable components and the price of the device but would also simplify and speed up the sign recognition process.

**Author Contributions:** J.P.: conceptualization, software, validation, formal analysis, investigation, resources, data curation, writing—original draft preparation, visualization. P.S.: methodology, supervision, writing—review and editing. All authors have read and agreed to the published version of the manuscript.

**Funding:** This research received no external funding.

**Institutional Review Board Statement:** The study was conducted according to the guidelines of the Declaration of Helsinki, and approved by the Institutional Ethics Committee of Technical University of Lodz (1/2021, date of approval: 28 October 2021).

**Informed Consent Statement:** Informed consent was obtained from all subjects involved in the study.

**Data Availability Statement:** The data presented in this article are publicly available at: https://figshare.com/search?q=10.6084%2Fm9.figshare.20304132 (accessed on 13 July 2022).

**Acknowledgments:** Jakub Piskozub would like to thank Lodz University of Technology for granting a scholarship.

**Conflicts of Interest:** The authors declare no conflict of interest.

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
