# Peer review of "Reducing the Number of Sensors in the Data Glove for Recognition of Static Hand Gestures"

_applsci, doi:10.3390/app12157388_

Round 1
Reviewer 1 Report
The authors present the article entitled “Reducing the number of sensors in the data glove for recognition of static hand gestures”.
This paper presents the advancement in electronic technology and machine learning algorithms through the report results of the study of designing a data glove equipped with piezoelectric sensors and validating a recognition task of hand gestures based on 16 static signs of the Polish Sign Language (PSL) alphabet.
The article presents the following concerns:
-
Feature application: I suggest removing this section. Please read the instructions for authors.
-
Introduction section is very extensive. I suggest to present subsection 1.1 as another section
-
.Figure 5 and 9: x-axis title is missing.
-
Include a table which compares the findings of the work vs the already reported in the stat of the art.
-
Line 270: Results?
-
I recommend giving an introduction between section 2 and 2.1.
-
I suggest adding a flowchart that describes the methodology.
-
Please add the equations used to calculate the performance.
-
Lines 335-336: Please improve the discussion of the table 2.
-
I recommend discussing Figure 9 in the Discussion section.
-
line 8, “Featured Application” is duplicated
-
The text must be written in the 3rd person or passive voice.
-
Add hyperlinks to tables, figures, and references.
-
The final part of the introduction should make a description of the structure of the text.
-
Line 54 can be justified with the BCI project reported in: A new approach for motor imagery classification based on sorted blind source separation, continuous wavelet transform, and convolutional neural network;
-
Before line 186, ANN should be introduced by talking about how they are used in different fields as: Neural network and spatial model to estimate sustainable transport demand in an extensive metropolitan area; Artificial neural networks in mppt algorithms for optimization of photovoltaic power systems: A review; Self-tuning neural network pid with dynamic response control
-
justify the relevance of this article with references from the journal.
-
The references must be according to the structure of the paper
The following misspelling should be checked:
-
line 15: It appears that you are missing a comma after the introductory phrase “In this paper”. Consider adding.
-
line 21: “shown…” Should be rewritten as“showed…”
-
line 24: “at a sustained…” Should be rewritten as “at sustained…” the indefinite article “a”, may be redundant when used with the uncountable noun “reliability” in your sentence.
-
line 28: It appears that you are missing a comma after the introductory phrase “In the last decade”. Consider adding.
-
line 196: It appears that “real time” is missing a hyphen. Consider adding: “real-time”.
-
line 248: “acquisition system was written in Python…” acquisition system written in Python…” The verb “was” appears to be unnecessary here.
-
line 252: The phrase “which enabled” may be wordy. consider changing by “enabling”.
-
line 323: The verb “archieve” is usually in the gerund form when following the word “allows”. Consider replacing it with the -ing form: “achieving”.
Author Response
Dear Reviewer,
We would like to thank for the valuable comments and suggestions, which helped us to improve the manuscript. Below is our point-by-point response to the concerns raised.
- The article presents the following concerns:
- Feature application: I suggest removing this section. Please read the instructions for authors.
This section has been removed as suggested.
- Introduction section is very extensive. I suggest to present subsection 1.1 as another section
As suggested, the Introduction section has been divided into two parts.
- Figure 5 and 9: x-axis title is missing.
The missing descriptions have been added.
- Include a table which compares the findings of the work vs the already reported in the stat of the art.
Table 3 comparing the studies has been added to the text of the article in the discussion section.
- Line 270: Results?
The template fragment was left in the text due to an oversight. It has been removed.
- I recommend giving an introduction between section 2 and 2.1.
A short introduction has been added between the sections
- I suggest adding a flowchart that describes the methodology.
We have added a flowchart in the new Fig. 5.
- Please add the equations used to calculate the performance.
An equation describing the method for calculating the classification efficiency of classifiers has been added.
- Lines 335-336: Please improve the discussion of the table 2.
The description of Table 2 has been expanded
- I recommend discussing Figure 9 in the Discussion section.
The description of figure 9 has been expanded
- line 8, “Featured Application” is duplicated
Duplication has been removed
- The text must be written in the 3rd person or passive voice.
The text has been rewritten in the passive voice.
- Add hyperlinks to tables, figures, and references.
Hyperlinks have been added to all tables, figures and references
- The final part of the introduction should make a description of the structure of the text.
At the end of the introduction, a brief description of the further content of the article has been added
- Line 54 can be justified with the BCI project reported in: A new approach for motor imagery classification based on sorted blind source separation, continuous wavelet transform, and convolutional neural network;
We have cited the suggested work.
- Before line 186, ANN should be introduced by talking about how they are used in different fields as: Neural network and spatial model to estimate sustainable transport demand in an extensive metropolitan area; Artificial neural networks in mppt algorithms for optimization of photovoltaic power systems: A review; Self-tuning neural network pid with dynamic response control
We have added a new citation on ANN no [21]. However, we found the above studies as not related to our work on the data glove. The above references deal with transport, photovoltaic energy, and dynamic response control.
- justify the relevance of this article with references from the journal.
Most articles published in Applied Sciences on sign language-based gesture recognition describe solutions based on vision-based methods. We included two articles in the body of this article: one describing a glove-based method that is similarly effective but has significant limitations to our solution [22]. The second article describes a vision-based method served as our justification for the limitations of such methods related to occlusion [13].
- The references must be according to the structure of the paper
References have been added according to the existing citation order
2. The following misspelling should be checked:
- line 15: It appears that you are missing a comma after the introductory phrase “In this paper”. Consider adding.
- line 21: “shown…” Should be rewritten as“showed…”
- line 24: “at a sustained…” Should be rewritten as “at sustained…” the indefinite article “a”, may be redundant when used with the uncountable noun “reliability” in your sentence.
- line 28: It appears that you are missing a comma after the introductory phrase “In the last decade”. Consider adding.
- line 196: It appears that “real time” is missing a hyphen. Consider adding: “real-time”.
- line 248: “acquisition system was written in Python…” acquisition system written in Python…” The verb “was” appears to be unnecessary here.
- line 252: The phrase “which enabled” may be wordy. consider changing by “enabling”.
- line 323: The verb “archieve” is usually in the gerund form when following the word “allows”. Consider replacing it with the -ing form: “achieving”.
We kindly thank you for pointing out the linguistic errors. All the above spelling errors have been corrected.
Reviewer 2 Report
1- The proposed method has been a frequently used study for a long time. In this respect, the superiority it provides over the alternatives or the contribution of the work should be well emphasized.
2- The difference between visual and wearable systems and why they are needed should be emphasized by citing to similar and competing studies.
3- The data of the volunteers were not discussed sufficiently. The distributions obtained during the measurements are not presented. Only the general results of the volunteers were shared.
4- Data and codes cannot be accessed. Please share at least for review and verification.
5- In the proposed method, it would be better if feature ranking methods such as relief, chi2, information gain or permutational feature selection were also applied and the findings were presented.
6- Findings Benchmarking with competitor/similar studies in terms of various parameters such as sensor number, sensor type, machine learning algorithm and performance metric will be important in terms of emphasizing the contribution.
Author Response
Dear Reviewer,
Thank you very much for the comments and suggestions. Below are our responses to the issues raised.
- The proposed method has been a frequently used study for a long time. In this respect, the superiority it provides over the alternatives or the contribution of the work should be well emphasized.
To best of our knowledge, no research team working on the data glove as a device for recognizing hand movements in sign language communication has addressed the topic of reducing the number of sensors used for this purpose. We have made it clear in the body of the article that this is the man contribution of our study not the construction of the glove itself. The added Table 3 comparing the results of the research teams presented in the Related Work section clearly shows that the other researchers assume the necessity of at least 5 sensors for hand gesture recognition, both dynamic and static. Despite the significantly reduced number of sensors, our results yield similar or even better recognition rates than those quoted. Moreover, we conducted the study on the second largest number of volunteers among all the cited works. As we pointed out in the introduction, the size of glove-translators is one of the main factors discouraging their potential use. We hope that the continuation of our research and the planned work on investigating the possibility of reducing the number of sensors required for effective hand gesture recognition will make it possible to significantly minimize the required hardware used for this purpose, thus potentially enabling their widespread use.
- The difference between visual and wearable systems and why they are needed should be emphasized by citing to similar and competing studies.
We have emphasized the shortcomings of the video systems in the Related work section in lines 142-146 and lines 193-199. The main disadvantages of the vision systems are defined by the developers of one such method described in the article [13]. We believe that the video we originally quoted as the source of our information does an excellent job of highlighting the advantages of direct methods over vision, so we want it to remain as one of the sources.
- The data of the volunteers were not discussed sufficiently. The distributions obtained during the measurements are not presented. Only the general results of the volunteers were shared.
Information on volunteers with confidentiality of this data has been completed. A data distribution chart has also been added in the new Fig. 6. We are aware of the number of outliers in the data set presented. Studies using the set with outliers removed yielded similar results in terms of feature selection () and better results in terms of classifier performance. Nevertheless, we decided to present the results obtained on the raw dataset recognizing that they better reflect the non-ideal performance of the device in real applications.
- Data and codes cannot be accessed. Please share at least for review and verification.
The data were transferred to another site allowing free access to them at 10.6084/m9.figshare.20304132. Data analysis was done by hand using the Matlab environment application Classification Learner. Therefore, there is no code that we can share for this part. The code controlling the glove has undergone a complete revision since the measurements were taken, and is now in a private repository at https://gitfront.io/r/user-8209221/aorBNVPYd81D/HandPi/ as handpi.py.
- In the proposed method, it would be better if feature ranking methods such as relief, chi2, information gain or permutational feature selection were also applied and the findings were presented.
Figure 7comparing the results of the predictor selection methods has been added to the body of the article. The chi2 and Kruskal Wallis methods did not yield conclusive results, so they are not included in the table.
- Findings Benchmarking with competitor/similar studies in terms of various parameters such as sensor number, sensor type, machine learning algorithm and performance metric will be important in terms of emphasizing the contribution.
Table 3 comparing the studies has been added to the text of the article in the discussion section.
Round 2
Reviewer 1 Report
The paper can be accepted.